# Influence of soil moisture levels on the growth and reproductive behaviour of *Avena fatua* and *Avena ludoviciana*

Sahil [1], Gulshan Mahajan[1]*, Deepak Loura[2], Katherine Raymont[3], Bhagirath Singh Chauhan[1,3]

**1** Queensland Alliance for Agriculture and Food Innovation (QAAFI), The University of Queensland, Gatton, Australia, **2** Chaudhary Charan Singh, Haryana Agricultural University, Hisar, India, **3** School of Agriculture and Food Sciences (SAFS), The University of Queensland, Gatton, Queensland, Australia

* g.mahajan@uq.edu.au

**Data Availability Statement:** All relevant data are within the manuscript and its Supporting Information files.

**Funding:** BSC received funding from the Grains Research and Development Corporation (GRDC)

## Abstract

Adaptation of weeds to water stress could result in the broader distribution, and make weed control task increasingly difficult. Therefore, a clear understanding of the biology of weeds under water stress could assist in the development of sustainable weed management strategies. *Avena fatua* (wild oat) and *A. ludoviciana* (sterile oat) are problematic weeds in Australian winter crops. The objectives of this study were to determine the growth and reproductive behaviour of *A. fatua* and *A. ludoviciana* at different soil moisture levels [20, 40, 60, 80, and 100% water holding capacity (WHC)]. Results revealed that *A. fatua* did not survive and failed to produce seeds at 20 and 40% WHC. However, *A. ludoviciana* survived at 40% WHC and produced 54 seeds plant$^{-1}$. *A. fatua* produced a higher number of seeds per plant than *A. ludoviciana* at 80 (474 vs 406 seeds plant$^{-1}$) and 100% WHC (480 vs 417 seeds plant$^{-1}$). Seed production for both species remained similar at 80 and 100% WHC; however, higher than 60% WHC. Seed production of *A. fatua* and *A. ludoviciana* was 235 and 282 seeds plant$^{-1}$, respectively, at 60% WHC. The 60% WHC reduced seed production of *A. fatua* and *A. ludoviciana* by 51 and 32% respectively, compared to 100% WHC. The plant height, leaf weight, stem weight, and root weight per plant of *A. fatua* at 60% WHC reduced by 45, 27, 32, and 59%, respectively, as compared with 100% WHC. Similarly, the plant height, leaf weight, stem weight, and root weight per plant of *A. ludoviciana* at 60% WHC reduced by 45, 35, 47 and 76%, respectively, as compared with 100% WHC. Results indicate that *A. ludoviciana* can survive and produce seeds at 40% of WHC, indicating the adaptation of the species to dryland conditions. The results also suggest that *A. ludoviciana* is likely to be robust under water stress conditions, potentially reducing crop yield. The ability of *A. fatua* and *A. ludoviciana* to produce seeds under water-stressed conditions (60% WHC) necessitates integrated weed management strategies that suppress these weeds whilst taking into account the efficient utilization of stored moisture for winter crops.

for investing in this research under project UA00084. The funders had no role in study design, data collection and analysis, decision to publish, or preparation of the manuscript.

**Competing interests:** The authors have declared that no competing interests exist.

## Introduction

Wild oat (weedy *Avena* spp) is a problematic and cosmopolitan weed in >20 crops across 55 countries and can cause an enormous yield loss in winter crops [1, 2]. Two wild oat species [i.e., *Avena fatua* L. (wild oats) and *A. ludoviciana* Durieu (sterile oats)] are the important weeds of this genus [3–7]. *A. fatua* and *A. ludoviciana* are very similar in morphological features and are difficult to differentiate at the vegetative stage; however, they can be easily differentiated at maturity. Seeds of *A. ludoviciana* shatter in pairs at plant maturity, while seeds of *A. fatua* shatter single [8].

Infestation of *A. fatua* and *A. ludoviciana* may cause yield reduction (~30–80%) in many winter crops such as wheat (*Triticum aestivum* L.), oat (*Avena sativa* L.), barley (*Hordeum vulgare* L.), rye (*Secale cereal* L.), pea (*Pisum sativum* L.), and canola (*Brassica napus* L.) [6, 9–16]. It has been reported that *A. fatua* and *A. ludoviciana* can cause ~70% yield loss in cereals [17]. The extent of yield reduction in these crops depends on the density and competitive ability of these species under different environmental conditions. In Australia, wild oat (*A. fatua* and *A. ludoviciana*) is ranked in the top three most important winter weeds, with an annual revenue loss due to infestation estimated at about AU$28 million [18]. *A. fatua* is highly problematic in southern and western Australia, while *A. ludoviciana* is more problematic in the northern grain region (NGR) of Australia [19, 20]. A survey reported that the geographical distribution of these species is increasing [21]. Therefore, it is very important to evaluate the biological factors and management practices that are responsible for a change in species distribution. It is likely that adaptation to water stress under changing climatic conditions could be one of the reasons for its wider distribution.

In Australia, *A. fatua* and *A. ludoviciana* have evolved resistance to ACCase and ALS inhibitor herbicides; therefore, poor control in crops is also a matter of great concern for crop production [22, 23]. Target and non-target-site herbicide-resistant mechanisms prevailed in these two species in Australia, potentially enhancing their further infestation [23]. These reasons suggest that knowledge of the biology of these weed species could aid in the development of better management practices. It is likely that intensive cropping systems and greater climatic variation in the NGR enabled the dominance of these species in the region. A modelling study of climate change in Australia revealed that rising temperatures would lead to frequent drought and fewer rainfall events [24]. Drought could play an important role in weed invasion by influencing crop-weed competition. In general, weeds flourished better than crops under water stress conditions due to inbuilt plasticity [25–27]. Adaptability in weeds for growth resources in response to crop competition might be due to biochemical or physiological changes in plants.

A recent study revealed that biotypes of *Sisymbrium thellungii* O. E. Schulz behaved differently under water stress conditions due to change in phenolics, proline and total soluble sugar contents [26]. This study also revealed that biotypes selected from different management environments differed in physiological behaviours such as stomatal conductance and net carbon assimilation [26]. Such information suggests that physiological and biochemical changes in plants in response to water stress are important mechanisms for drought avoidance [28,29].

*Avena ludoviciana* has a greater tendency to adapt to water stress as compared to *A. fatua* [30]; however, current information is limited to seed germination studies conducted in the laboratory. In the future, the crop-weed competition will be increasingly influenced by water stress, presenting the potential to reduce crop yield and promote weed invasiveness where particular weed species grow more robustly. The situation could become more serious if such weeds have the capability to produce sufficient seeds under these conditions [31]. A recent study in Australia revealed that *S. thellungii* could produce ~2000 seeds per plant under 25%

WHC of the soil [26]. Similarly, *Echinohloa colona* (L.) Link produced sufficient seeds (~8000 seeds per plant) for reinfestation under water stress conditions (25% WHC). These studies also showed that phenological, physiological, and biochemical changes in weed species are variable in response to water stress and that these are highly affected by both the species and degree of water stress. It has been reported that *A. fatua* attained lower biomass at reduced soil moisture levels as compared to the field capacity; however, no information on weed seed production was provided in this study [32]. Therefore, studies on the growth and reproductive behaviour of *A. fatua* and *A. ludoviciana* in response to water stress are essential for generating effective weed management strategies. Compared to other Australian weed species, information on *A. fatua* and *A. ludoviciana* in response to water stress is limited. Therefore, this study was undertaken to better understand (1) how growth and reproductive behaviour of *A. fatua* and *A. ludoviciana* change in response to water stress, and (2) how chemical changes influence the reproductive potential of *A. fatua* and *A. ludoviciana*.

## Material and methods

### Experiment set up and conditions

To run this experiment, the permission was taken from the property owner for seed collection, while for subsequent field and screenhouse experiments, the permission was taken from The University of Queensland, Gatton. The experiment was conducted in a randomized complete block design with 10 treatments [two species of wild oat (*A. fatua* and *A. ludoviciana*) and five soil moisture levels (100%, 80%, 60%, 40% and 20% WHC, simulating different degrees of water stress: no, light, mild, high, and severe water stress, respectively) replicated eight times in factorial arrangement. A total of 80 pots (7.5 L volume) were used in the experiment. Each pot was filled with 3.4 kg of soil mixture (2:1 mixture of field soil and potting mix). The soil was mixed with potting mix to provide better aeration and moisture to the plants. Four seeds were sown in each pot and after establishment, seedlings were thinned to a single plant. Out of eight replications, five replications were used for assessing growth and reproductive behaviour and the remaining three replications were used for biochemical studies.

Seeds of *A. fatua* and *A. ludoviciana* used in this study were collected from Warialda (GPS coordinates: latitude -29.395 and longitude, 150.620), New South Wales, with the permission of property owner in October 2017 and multiplied at the Research Farm of the University of Queensland, Gatton in the winter season of 2018. Collection of seeds was made from 50–60 matured mother plants. Seeds were stored in the laboratory at room temperature until used in the experiment. The experiment started on 4 May 2019 in the screen house (a structure with overhead transparent polyethylene cover to prevent rain damage) of the University of Queensland and was terminated on 3 October 2019. The water stress treatments were imposed on 16 June 2019 and continued until 7 days before harvest. No fertilizer was applied to the pots.

For determining the WHC, a modified method used by previous workers was employed [33]. Three pots were filled with 3.4 kg soil mixture (2:1 mixture of field soil and potting mix) and saturated with tap water. The pots were then covered with a black plastic cover and allowed to drain for 48 h. Thereafter, the plastic covers were removed, and 300 g potting mix samples were taken from the middle of each pot. These samples were weighed (wet weight of potting mix, $X$) before being oven-dried (90 ˚C for 72 h) and reweighed after drying (dry weight of potting mix, $Y$). The WHC was then calculated using the formula $(X-Y) \times 100/Y$. The 80, 60, 40, and 20% WHC levels were determined as a fraction of the 100% WHC found. The 100% WHC of the soil mixture was 1.8 L per pot. To maintain the WHC during the study, each pot was weighed and a measurable quantity of tap water was slowly added over the

surface of the soil on alternating days. Extra plants were grown to account for the weight of growing plants and used to determine plant weight grown in different treatments.

## Sampling and measurements

Plant height, tiller number per plant, leaf number per plant, and chlorophyll content (SPAD readings) were determined at a 7-day interval starting 44 days after planting (DAP) until harvest. Plant height in each pot was measured from the soil surface to the uppermost tip of the plant. To measure chlorophyll content in leaves, a chlorophyll meter, SPAD-502 (Soil-Plant Analyses Development, Konica Minolta Sensing, Inc, Japan: Model number 71923020) was used. The top three leaves from each plant were selected to measure the chlorophyll content (SPAD readings).

At harvest, panicles and seeds per panicle were counted for individual plants. The total number of shattered seeds were also counted for individual plants. The aboveground shoot, leaf, and root biomass were also determined separately at harvest. Samples were placed in paper bags and dried in an oven at 70 ˚C for 72 h and then weighed. Daily maximum and minimum temperatures were recorded using a Tinytag data logger (Gemini data loggers, Chichester, UK) installed in the screen house. The maximum and minimum temperatures in the screen house varied from 17.3–45.5 ˚C and 1.4 to 15.1 ˚C, respectively (Fig 1). The minimum temperature was relatively lower from mid-June to mid-July and the maximum temperature was relatively higher in September and October.

## Biochemical parameters

Undamaged, fresh, and healthy leaves (ca. 5 g) from each plant were collected at 65 DAP. The samples were stored in zipped lock plastic bags at 4 ˚C until used for analysis (i.e., 10 days). From these leaf samples, soluble phenolics were determined by following the procedure of previous workers [34]. The total soluble sugar content of each sample was determined by following the phenol sulphuric method [35] and improved by various workers [36]. The free proline content of each sample was measured by the following method of previous workers [37].

**Statistical analyses.**   Data were subjected to analysis of variance (ANOVA) by using the software Elementary Designs Application 1.0 Beta (AgriStudy.com: www.agristudy.com) (verified with GEN STAT 16th Edition; VSN International, Hemel Hempstead, UK). All data met assumptions of normality of residuals and homogeneity of variance. Means were separated using a Fisher's protected least significant difference (LSD) test at $P = 0.05$ (see S1 Data).

Growth data were analyzed using regression analysis for plant height, tiller numbers, and leaf numbers per plant.

$$y = a/\{1 + \exp[-(x - d50)/b]\}$$

where $y$ is the estimated plant height or tiller number per plant at time $x$ (DAP); $a$ is the maximum of the parameter; $d50$ is the time to reach 50% of the final plant height or tiller number per plant, and $b$ is the slope.

## Results and discussion

### Morphological and growth parameters

Plant height of *A. fatua* followed a sigmoid pattern with a maximum height of 78 and 108 cm at 150 DAP at 80 and 100% WHC, respectively (Fig 2a). Plant height of *A. ludoviciana* also followed a sigmoid pattern with a maximum height of 55, 66, and 101 cm at 150 DAP at 60, 80, and 100% WHC, respectively (Fig 3a). At 100% WHC, *A. fatua* and *A. ludoviciana* took 135

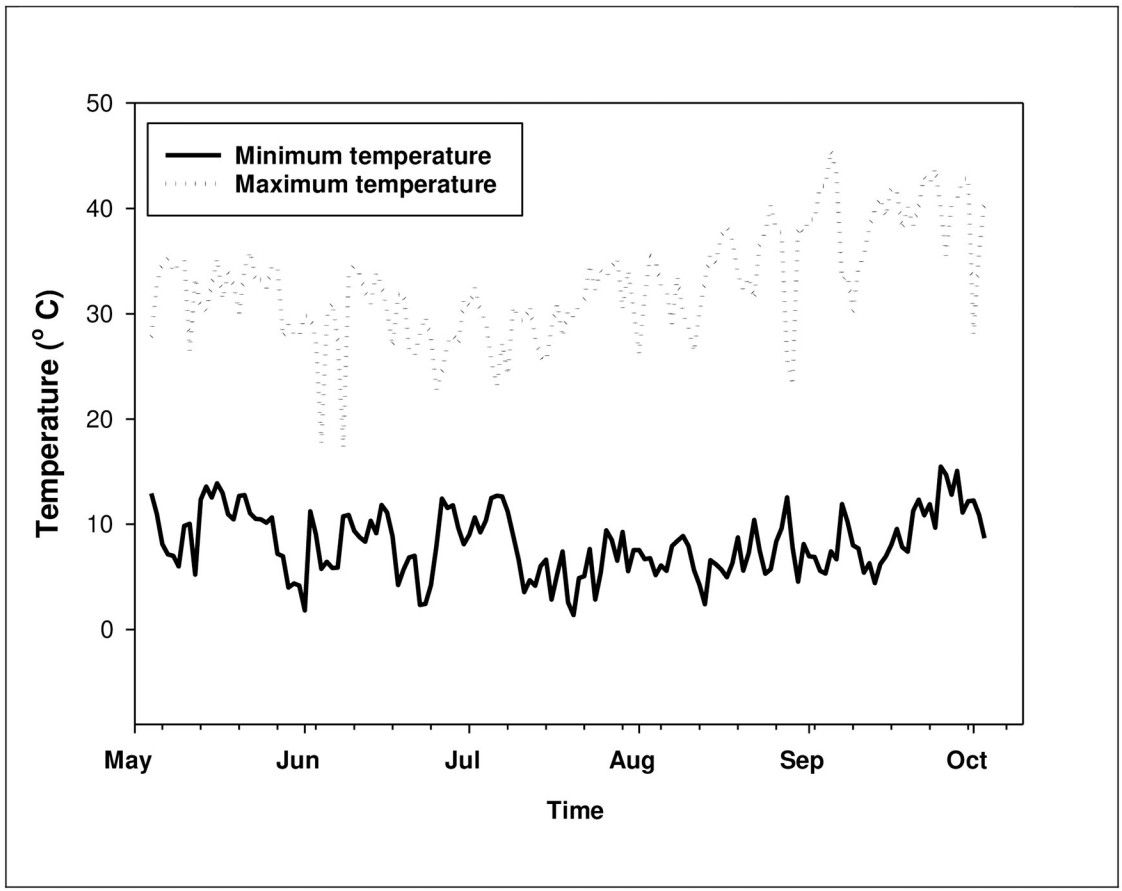

**Fig 1. Maximum and minimum temperature (˚ C) in the screen house during the experimental study (2019).**

and 148 days to attain 50% height (Table 1 and Fig 1a). *A. fatua* plants died at 107 and 121 DAP at 20 and 40% WHC, respectively, (extreme water stress conditions). At 120 DAP, *A. fatua* attained 46, 70, and 80 cm height at 60, 80, and 100% WHC, respectively. Unlike *A. fatua*, *A. ludoviciana* plants survived at 40% WHC and the maximum height was 44 cm at 150 DAP. Plants of *A. ludoviciana* at 20% WHC died at 113 DAP due to scarcity of water, suggesting that the plants would not survive in extreme water stress conditions. At 120 DAP, plants of *A. ludoviciana* attained 39, 50, 65, and 66 cm height at 40, 60, 80, and 100% WHC levels, respectively. Similar results were obtained for *E. colona* in response to water stress in Australia, where maximum and minimum height were 90 and 49 cm at 100 and 25% WHC, respectively [27].

Both *A. fatua* and *A. ludoviciana* attained greater height at 100% WHC compared to 75% WHC (Table 2), suggesting that both species grew with full potential at 100% WHC for maximum plant growth. In water stress situations in the field, winter crops such as wheat and barley may suffer due to scarcity of water, and *A. fatua* and *A. ludoviciana* may grow with superior vigour and competitiveness. In such situations, it is important to control early cohorts of *A. fatua* and *A. ludoviciana* and save soil moisture for crops.

Tiller production of *A. ludoviciana* at 60% WHC reduced by 24% compared with 100% WHC; however, tiller production of *A. fatua* remained similar at 60 and 100% WHC (Table 2). Contrary to this, leaf production of *A. ludoviciana* did not decrease at 60 and 100% WHC, while leaf production of *A. fatua* decreased by 65% at 60% WHC compared with 100%

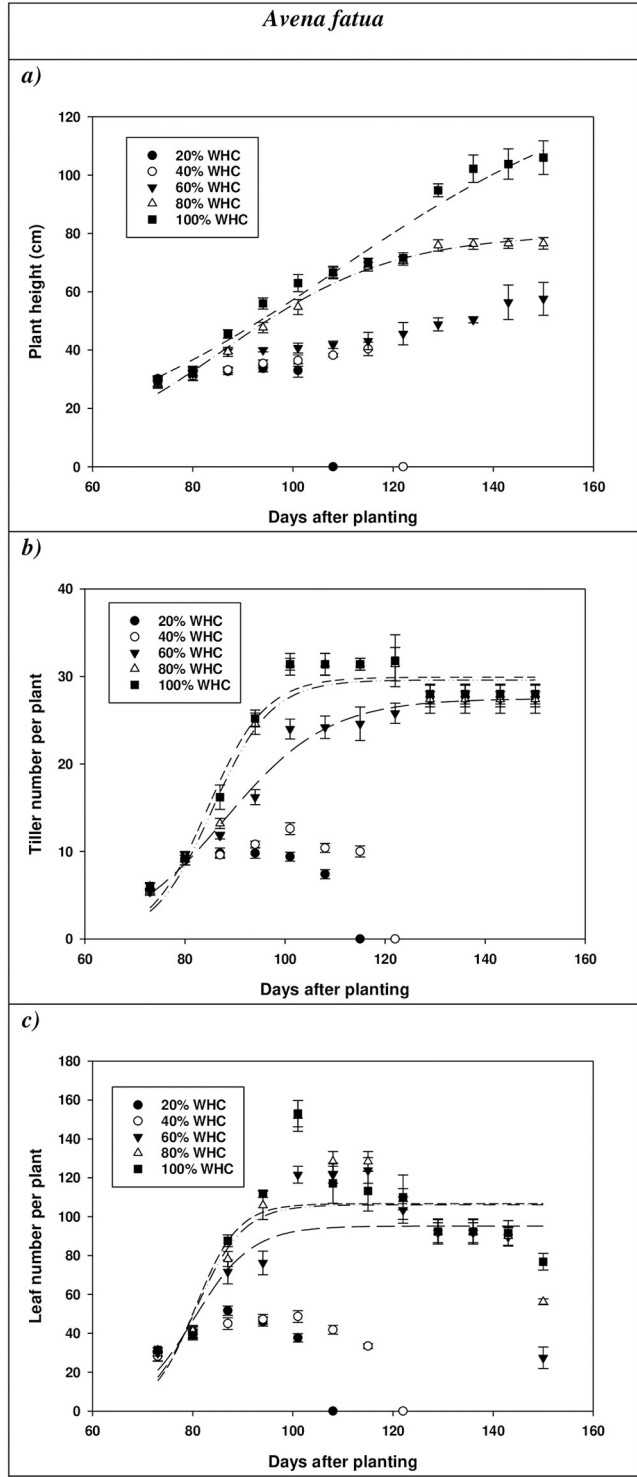

**Fig 2. a) Plant height, b) tiller number and c) leaf number per plant of *Avena fatua* at various stages of plant growth.**

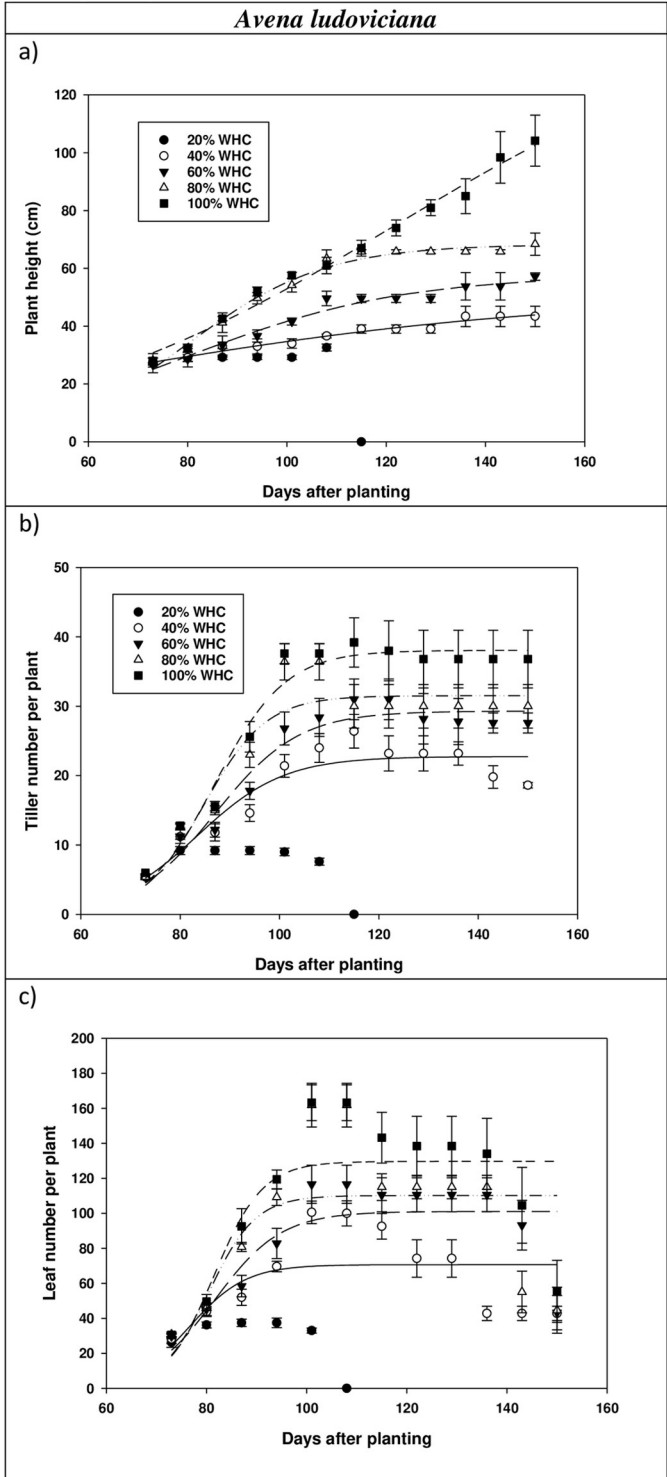

**Fig 3. a) Plant height, b) tiller number and c) leaf number per plant of *Avena ludoviciana* at various stages of plant growth.**

**Table 1. Regression parameters of a three-parameter sigmoid model[a]** ($y = a/\{1 + \exp[-(x-d50)/b]\}$) **fitted to** *Avena fatua* **and** *Avena ludoviciana* **plant height, tiller and leaf number per plant at different soil moisture levels.**

| Species and soil moisture levels (% WHC) | Plant height (cm) | | | | Tiller number per plant | | | | Leaf number per plant | | | |
|---|---|---|---|---|---|---|---|---|---|---|---|---|
| | $a \pm$ SE | $b \pm$ SE | $d50 \pm$ SE | $R^2$ | $a \pm$ SE | $b \pm$ SE | $d50 \pm$ SE | $R^2$ | $a \pm$ SE | $b \pm$ SE | $d50 \pm$ SE | $R^2$ |
| *A. ludoviciana* | | | | | | | | | | | | |
| 20 | Curve did not fit | | | | Curve did not fit | | | | Curve did not fit | | | |
| 40 | 44.5 ± 12.1 | 64.4 ± 6.0 | 50.3 ± 4.7 | 0.98 | 8.7 ± 2.8 | 83.5 ± 2.7 | 22.7 ± 1,2 | 0.84 | 5.6 ± 5.2 | 76.6 ± 5.7 | 70.6 ± 8.2 | 0.30 |
| 60 | 22.3 ± 3.7 | 78.9 ± 2.2 | 57.8 ± 2.4 | 0.97 | 8.3 ± 1.6 | 87.1 ± 1.7 | 29.3 ± 1.0 | 0.95 | 6.7 ± 4.2 | 81.6 ± 4.2 | 101.1 ± 8.9 | 0.60 |
| 80 | 13.7 ± 1.3 | 80.2 ± 1.0 | 68.3 ± 1.2 | 0.96 | 6.3 ± 1.9 | 84.7 ± 2.1 | 31.5 ± 1.4 | 0.94 | 4.6 ± 4.5 | 80.5 ± 5.1 | 110.2 ± 13.6 | 0.39 |
| 100 | 35.9 ± 5.6 | 12.1 ± 12.0 | 148.5 ± 24.4 | 0.98 | 7.1 ± 1.1 | 87.1 ± 1.2 | 38.0 ± 1.0 | 0.96 | 4.8 ± 3.4 | 81.5 ± 3.7 | 129.0 ± 11.5 | 0.59 |
| *A. fatua* | | | | | | | | | | | | |
| 20 | Curve did not fit | | | | Curve did not fit | | | | Curve did not fit | | | |
| 40 | Curve did not fit | | | | Curve did not fit | | | | Curve did not fit | | | |
| 60 | Curve did not fit | | | | 102.0 ± 1.1 | 88.0 ± 1.1 | 27.5 ± 0.6 | 0.98 | 5.7 ± 5.0 | 80.2 ± 5.1 | 95.2 ± 10.6 | 0.44 |
| 80 | 16.8 ± 1.5 | 86.1 ± 1.2 | 79.9 ± 1.8 | 29.6 ± 1.1 | 5.8 ± 1.4 | 85.5 ± 1.6 | 29.6 ± 1.1 | 0.92 | 4.7 ± 3.6 | 80.7 ± 3.9 | 106.1 ± 9.7 | 0.43 |
| 100 | 29.3 ± 6.4 | 109.0 ± 10.9 | 135.3 ± 21.7 | 29.9 ± 0.8 | 5.8 ± 1.1 | 84.5 ± 1.2 | 29.9 ± 0.8 | 0.95 | 4.3 ± 2.7 | 80.5 ± 2.9 | 106.7 ± 4.7 | 0.64 |

In the model, *y* is the estimated plant height or tiller or leaf number per plant at time x (days after planting) *a* is the maximum of the parameter; *d50* is the time to reach 50% of the final plant height or tiller or leaf number per plant; and *b* is the slope; SE: standard error (±).

WHC (Table 2). In a similar study, higher number of leaves in *S. thellungii* (27 leaves plant[-1]) were reported at 100% WHC compared with the soil moisture level of 25% WHC (20 leaves plant[-1]) [26]. Similarly, another study observed 52 leaves plant[-1] for *Lactuca serriola* L. at 100% WHC as compared with 42 leaves plant[-1] at 50% WHC. At 100% WHC, *A. fatua* and *A. ludoviciana* plants took 38 and 30 days for 50% tillering (Table 1; Figs 2b and 3b) [38]. For *A. fatua* and *A. ludoviciana*, the number of leaves per plant followed a similar sigmoidal trend at 60, 80 and 100% WHC (Figs 2c and 3c). At 100% WHC, *A. fatua* and *A. ludoviciana* plants took 129 and 107 days to produce 50% leaves per plant (Table 1).

For *A. ludoviciana*, leaf weight, stem weight, and root weight per plant increased with the increased soil moisture level up to 100% WHC (Table 3). However, in case of *A. fatua*, only

**Table 2. Morphological and growth parameters of** *Avena fatua* **and** *Avena ludoviciana* **in relation to soil moisture levels at harvest.**

| Species and parameters | Soil moisture levels (% WHC) | | | | |
|---|---|---|---|---|---|
| | 20 | 40 | 60 | 80 | 100 |
| **Plant height (cm plant[-1])** | | | | | |
| *A. ludoviciana* | 0a | 43.4b | 57.2c | 68.4cd | 104.2e |
| *A. fatua* | 0a | 0a | 57.7c | 76.6d | 106.0e |
| LSD (0.05) | 12.3 | | | | |
| **Tiller (number plant[-1])** | | | | | |
| *A. ludoviciana* | 0a | 19b | 28c | 30c | 37d |
| *A. fatua* | 0a | 0a | 27c | 27c | 28c |
| LSD (0.05) | 5.1 | | | | |
| **Leaf (number plant[-1])** | | | | | |
| *A. ludoviciana* | 0a | 43bc | 43bc | 45bc | 54c |
| *A. fatua* | 0a | 0a | 27b | 56c | 78d |
| LSD (0.05) | 23.0 | | | | |

WHC: water holding capacity; LSD: least significant differences. Any two values in the same column followed by the same letter are not significantly different ($P = 0.05$).

**Table 3. Chlorophyll content (SPAD reading), leaf weight, stem weight and root weight per plant of *Avena ludoviciana* and *Avena fatua* in relation to soil moisture levels at harvest.**

| Species and parameters | Soil moisture levels (% WHC) | | | | |
|---|---|---|---|---|---|
| | 20 | 40 | 60 | 80 | 100 |
| Chlorophyll content (SPAD) value | | | | | |
| *A. ludoviciana* | 0a | 25.2bc | 30.4c | 36.4de | 42.8e |
| *A. fatua* | 0a | 0a | 19.5b | 33.5cd | 42.0e |
| LSD (0.05) | 7.8 | | | | |
| Leaf weight (g plant⁻¹) | | | | | |
| *A. ludoviciana* | 0a | 5.3b | 6.3b | 6.4b | 9.7d |
| *A. fatua* | 0a | 0a | 6.5b | 8.0c | 8.9cd |
| LSD (0.05) | 1.3 | | | | |
| Stem weight (g plant⁻¹) | | | | | |
| *A. ludoviciana* | 0a | 3.9b | 21.1d | 31.1e | 39.8f |
| *A. fatua* | 0a | 0a | 14.5c | 31.5e | 46.2g |
| LSD (0.05) | 3.5 | | | | |
| Root weight (g plant⁻¹) | | | | | |
| *A. ludoviciana* | 0a | 4.2b | 6.9bc | 14.7c | 25.9d |
| *A. fatua* | 0a | 0a | 4.7b | 9.3bc | 11.4c |
| LSD (0.05) | 6.6 | | | | |

WHC: water holding capacity; LSD: least significant differences. Any two values in the same column followed by the same letter are not significantly different (*P* = 0.05).

stem weight per plant increased with increasing soil moisture levels up to 100% WHC. At 60% WHC, *A. fatua* had lower stem weight per plant than *A. ludoviciana*. A similar trend was noticed for chlorophyll content (SPAD readings) at 60% WHC. A previous study observed 55, 57, and 38% reduction in the leaf area, dry weight, and the number of viable tillers of *A. fatua*, respectively, with a decreasing soil moisture regime from 2.5 to 0.6 cm water per week [32]. A recent water stress study on *E. colona* in Australia revealed that some biotypes of *E. colona* had higher root and shoot biomass at 100% WHC as compared with 75% WHC [27].

## Biochemical attributes

Soil moisture had a significant effect on leaf biochemistry of *A. fatua* and *A. ludoviciana* at 65 DAP (Table 4). Both species behaved similarly with varying soil moisture levels. Averaged over

**Table 4. Biochemical attributes in relation to soil moisture levels at 65 days after planting (similar behaviour for both species was observed; therefore, observations were averaged).**

| Soil moisture levels (% WHC) | Total soluble phenolics (mg g⁻¹ of fresh weight) | Free proline (mg g⁻¹ of fresh weight) | Total soluble sugar (mg g⁻¹ of fresh weight) |
|---|---|---|---|
| 20 | 3.22c | 48.2c | 6.33d |
| 40 | 2.72b | 15.9b | 4.15bc |
| 60 | 2.46b | 2.02ab | 3.11b |
| 80 | 2.03a | 0.43a | 1.99a |
| 100 | 2.02a | 0.26a | 2.19ab |
| LSD (0.05) | 0.35 | 15.4 | 1.10 |

WHC: water holding capacity; LSD: least significant differences. Any two values in the same column followed by the same letter are not significantly different (*P* = 0.05).

weed species, the amount of total soluble phenolics, free proline, and soluble sugars was higher at 20 and 40% WHC compared with 100% WHC. At 60% WHC, total soluble phenolics increased by 21% compared with 100% WHC.

This increase in soluble sugars in leaves under 20% WHC counteracted the osmotic stress; however, *A. fatua* plants could not survive at 20 and 40% WHC. The antioxidant action of phenolic compounds that played an essential role in neutralizing free radicals increased in *A. fatua* and *A. ludoviciana* at 20 and 40% WHC; however, it could not help *A. fatua* plants to survive under severe water stress conditions. Recent studies on winter weeds in Australia revealed that the amount of total soluble phenolics, free proline, and soluble sugars increased with an increase in water stress [26, 39].

## Seeds and panicles plant[-1]

Seed production of *A. fatua* and *A. ludoviciana* decreased with increasing soil moisture stress and at 20% WHC, both species failed to produce seeds (Table 5). The 60% WHC reduced seed production of *A. fatua* and *A. ludoviciana* by 51 and 32%, respectively, compared to 100% WHC (Table 5). However, the seed production of *A. fatua* and *A. ludoviciana* did not differ between 80 and 100% WHC. Panicle numbers of *A. fatua* decreased by 55% at 60% WHC compared with 100% WHC, but *A. ludoviciana* panicle number remained similar at 60 and 100% WHC. At 40% WHC, *A. fatua* did not produce seeds; however, *A. ludoviciana* still produced 54 seeds plant[-1]. It was also observed that at 80 and 100% WHC, *A. fatua* had a higher number of seeds plant[-1] than *A. ludoviciana*.

Projected climate change in Australia has revealed that drought spells may increase in the future, and how crops and weeds respond to the water stress environment remains a priority for weed management [24]. In our study, *A. fatua* and *A. ludoviciana* demonstrated greatest growth and highest reproductive potential at 80 and 100% WHC. Higher seed production of *A. fatua* and *A. ludoviciana* at 80 or 100% WHC compared with 60% WHC was attributed to increased stem and root weight. *A. ludoviciana* was also able to grow and produce sufficient seeds (54 seeds plant[-1]) even at 40% WHC, indicating that *A. ludoviciana* has a greater potential for proliferation than *A. fatua* under drought conditions. At 60% WHC, *A. fatua* had a lower number of leaves per plant than *A. ludoviciana*, indicating that photosynthetic efficiency of *A. fatua* decreased with increasing water stress and *A. fatua* plants could not survive at 40 and 20% WHC due to photosynthesis limited by the reduced leaf number. This was also

**Table 5. Panicles per plant and seeds per plant of *Avena ludoviciana* and *Avena fatua* in relation to soil moisture levels.**

| Species and parameters | Soil moisture levels (% WHC) | | | | |
|---|---|---|---|---|---|
| | **20** | **40** | **60** | **80** | **100** |
| Panicles (number plant[-1]) | | | | | |
| *A. ludoviciana* | 0a | 4a | 17b | 19b | 20b |
| *A. fatua* | 0a | 0a | 19b | 33c | 42d |
| LSD (0.05) | 7 | | | | |
| Seeds (number plant[-1]) | | | | | |
| *A. ludoviciana* | 0a | 54b | 282c | 406d | 417d |
| *A. fatua* | 0a | 0a | 235c | 474e | 480e |
| LSD (0.05) | 57 | | | | |

WHC: water holding capacity; LSD: least significant differences. Any two values in the same column followed by the same letter are not significantly different (*P* = 0.05).

evidenced by the lower number of leaves of *A. fatua* (42 leaves per plant$^{-1}$) plants as compared with *A. ludoviciana* (70 leaves plant$^{-1}$) at 100 DAP (Figs 2c and 3c).

The current study suggests that *A. ludoviciana* can survive at WHC as low as 40%, while maintaining sufficient seed production for reinfestation. The basis for adaptation of *A. ludoviciana* to water stress is reflective of the increased leaf number and chlorophyll content (SPAD reading) as compared with *A. fatua* under water stress (60% WHC). These observations suggest that under a drought environment, *A. ludoviciana* may expand its range due to its ability to tolerate water stress. Under severe water stress (20% WHC), *A. fatua* and *A. ludoviciana* could not survive, suggesting that sufficient soil moisture is necessary for their reproduction.

The increase in plant height of *A. fatua* and *A. ludoviciana* at 100% WHC compared with lower soil moisture levels (60% WHC) could be a result of the increased cell enlargement due to high turgor pressure at high soil moisture conditions [39]. Many weeds showed reduced shoot height when seedlings were exposed to drought [33, 40]. A stronger root system of *A. ludoviciana* and *A. fatua* at 100% WHC as compared with 60% WHC indicates their ability to efficiently utilize soil moisture for high seed production [41]. Thus, plants with high root biomass could efficiently take soil nutrients and water, while those with a higher proportion of stem and leaf weight per plant can collect more light energy [42].

The ability to produce seeds under a water stress environment could play an important role in influencing weed population dynamics in the wake of climate change. The most sustainable weed control approach is to reduce the weed seed bank in the soil [43, 44]. *A. fatua* did not produce seeds at 40% WHC, while a small amount of *A. ludoviciana* seeds (54 seeds plant$^{-1}$) at 40% WHC pose the potential for the infestation in the succeeding crop. In mild water stress conditions (i.e., 60% WHC), both species can produce sufficient seeds for reinfestation, and they may also result in decreased crop yield due to increased crop-weed competition as a result of reduced crop growth [45].

The depletion of weed seedbank is a viable long-term strategy for weed control [44]. This can be achieved by harvest weed seed control practices or utilising herbicide application to kill weeds at an early stage. Harvest weed seed control practices can become important in a no-till water-stressed field where pre-emergent herbicides are less effective and stressed weeds are difficult to control with post-emergent herbicides due to reduced herbicide absorption and physiological activity of weeds [46]. Significant reductions in *A. fatua* and *A. ludoviciana* populations can also be achieved through cultural practices, such as increased crop densities or closer row spacing. However, these practices should be aligned to agronomic practices developed under drought-research programs. In a previous study, the growth, biomass, and seed production potential of *A. fatua* and *A. ludoviciana* were reduced by an increased wheat planting density in the field [47]. In another study, an increased rate of wheat seed helped in reducing the seed number of *A. fatua* [6].

In water-scarce environments, *A. fatua* and *A. ludoviciana* growth might be reduced by using drought-tolerant crop cultivars or weed-competitive crop cultivars [48] or by manipulating crop sowing time [3, 49]. Management strategies should target for control of early cohorts of weeds in crops. The effect of enhanced weed competition under drought environments could severely harm crop yield; therefore, it is important to control early cohorts of weeds, so that stored moisture can be used for crops.

## Conclusions

The results of this study indicated that *A. fatua* and *A. ludoviciana* attained maximum plant height, seed number, and stem and root biomass at 100% WHC compared with 60% WHC. At

60% WHC, there was a reduction in the growth parameters and the number of seeds per plant. *A. fatua* was more sensitive to water stress because, at 20 and 40% WHC, plants were not able to survive. However, at 40% WHC, *A. ludoviciana* plants survived and produced seeds. A considerable reduction in *A. fatua* and *A. ludoviciana* growth and seed production was observed at 60% WHC as compared with 100% WHC. Since the main winter crops, such as wheat, require >60% WHC to avoid yield reduction, growers implement favourable soil moisture regimes throughout the growing season in irrigated wheat and barley crops [50]. In such situations, if *A. fatua* and *A. ludoviciana* plants are not controlled, eventually they will optimally grow, reproduce, and interfere with crops due to the available soil moisture. The high reproductive ability of *A. fatua* and *A. ludoviciana* at favourable soil moisture conditions indicate that their high rates of infestation may occur in irrigated crops. The ability of *A. ludoviciana* to produce seeds under high water stress conditions could increase its invasiveness throughout Australia in fallow and dryland conditions owing to its high adaptation. Future research needs to be evaluated on exploring integrated weed management tools to enhance the competitiveness of wheat and barley for the suppression of *A. fatua* and *A. ludoviciana* under favourable soil moisture and drought conditions.

## Supporting information

**S1 Data. Data and statistical analysis information.**
(XLSX)

## Author Contributions

**Conceptualization:** Gulshan Mahajan, Bhagirath Singh Chauhan.

**Data curation:** Sahil, Deepak Loura.

**Formal analysis:** Sahil, Gulshan Mahajan.

**Funding acquisition:** Bhagirath Singh Chauhan.

**Methodology:** Sahil, Gulshan Mahajan, Katherine Raymont, Bhagirath Singh Chauhan.

**Resources:** Bhagirath Singh Chauhan.

**Supervision:** Katherine Raymont, Bhagirath Singh Chauhan.

**Writing – original draft:** Sahil.

**Writing – review & editing:** Gulshan Mahajan, Deepak Loura, Katherine Raymont, Bhagirath Singh Chauhan.

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
