## [Decision Letter · Decision Letter 0]

18 May 2020

PONE-D-20-06038

Influence of soil moisture levels on the growth and reproductive  behaviour of Avena fatua and Avena ludoviciana

PLOS ONE

Dear Dr. Mahajan,

Thank you for submitting your manuscript to PLOS ONE. After careful consideration, we feel that it has merit but does not fully meet PLOS ONE’s publication criteria as it currently stands. Therefore, we invite you to submit a revised version of the manuscript that addresses the points raised during the review process.

I have received 2 reports for your submission. Both referees have positive views on the manuscript and think that the work is worth publishing. However, both have suggested several minor changes.Both reviewers are concerned with the repeated use of Latin names of both species, although common names vary from region to region, differentiate these both species by their common names in Australia.More information is needed on seed collection. Whether the seeds were collected from single population? How many mother plants were there? Weed species exhibit inter and intra population differences.The current version is not formatted according to Plos One.Please give due attention to all comments while revising your manuscript. I look forward to receive your revised submission.

We would appreciate receiving your revised manuscript by Jul 02 2020 11:59PM. To enhance the reproducibility of your results, we recommend that if applicable you deposit your laboratory protocols in protocols.io, where a protocol can be assigned its own identifier (DOI) such that it can be cited independently in the future. For instructions see: http://journals.plos.org/plosone/s/submission-guidelines#loc-laboratory-protocols

We look forward to receiving your revised manuscript.

Kind regards,

Shahid Farooq, Ph.D.

Academic Editor

PLOS ONE

Reviewers' comments:

Reviewer's Responses to Questions

**Comments to the Author**

1. Is the manuscript technically sound, and do the data support the conclusions?

Reviewer #1: Yes

Reviewer #2: Yes

2. Has the statistical analysis been performed appropriately and rigorously? 

Reviewer #1: Yes

Reviewer #2: Yes

3. Have the authors made all data underlying the findings in their manuscript fully available?

Reviewer #1: Yes

Reviewer #2: Yes

4. Is the manuscript presented in an intelligible fashion and written in standard English?

Reviewer #1: Yes

Reviewer #2: Yes

5. Review Comments to the Author

Reviewer #1: Overall, the manuscript entitled “Influence of soil moisture levels on the growth and reproductive behaviour of Avena fatua and Avena ludoviciana” presents novel and relevant results regarding the ability of these species in withstanding dryer soil conditions, a condition which is expected to become increasingly more common in the future due to ongoing global climate change.

Comment #1: It seems no information is presented regarding the actual cycle length across soil moisture levels. Is such information available?

Comment #2: figures seem to be lacking quality, appearing blurred.

Line 52: scientific names of two species which are referred to as “wild oats” throughout the manuscript had already been properly introduced (line 41-42); why inform their scientific names again? This can be confusing to readers.

Line 121: this statement regarding the actual ability of crops such as wheat and barley seems vague and is not supported by any literature citation. It is clear that the authors are simply indicating this possibility, but it would be better to refer to articles (if any) which assessed these crops´ growth under water deficit. If none is found, then it seems necessary to add a sentence that “such statement remains to be evaluated” or similar.

Reviewer #2: I have evaluated the manuscript "Influence of soil moisture levels on the growth and reproductive behaviour of Avena fatua and Avena ludoviciana" written by Sahil et al. Overall, it is an excellent piece of study and deserves publication.

I have a few minor suggestions (commented on the attached manuscript file).

The opening sentence of abstract defines climate change and study is on water stress. Only water stress can not be considered as climate change. Please rewrite the sentence.

The Methods are given after results, while plos one requires methods before results.

Results and discussion are combined. Could these be separated?

More info is needed on seed collection and storage. I suggest to add coordinates or a map showing the locations.

Authors have cited a survey work in Australia, could the current distribution map be provided in the manuscript?

I suggest to use common names of both species, how these are separated in Australia? The use of Latin names have been redundant throughout the manuscript. Obviously, common names will be redundantly used but better to use.

6. PLOS authors have the option to publish the peer review history of their article (what does this mean?). If published, this will include your full peer review and any attached files.

Reviewer #1: No

Reviewer #2: No

---

## [Author Response · Author response to Decision Letter 0]

26 May 2020

Editor comments

• I have received 2 reports for your submission. Both referees have positive views on the manuscript and think that the work is worth publishing. However, both have suggested several minor changes.

Thanks, the suggested changes have been incorporated.

• Both reviewers are concerned with the repeated use of Latin names of both species, although common names vary from region to region, differentiate these both species by their common names in Australia.

In Australia, both are known as black or wild oats. Therefore, we have incorpared common names used in the USA (WSSA). Also, it will be difficult to understand by readers if we use common names throughout the manuscript. Therefore, we have used Latin names throughout the text.

• More information is needed on seed collection. Whether the seeds were collected from single population? How many mother plants were there? Weed species exhibit inter and intra population differences.

The initial seeds were collected from a farmer’s field. These seeds were grown at the Gatton research farm in the next season and fresh seeds collected from those 50-60 matured plants were used for the study. 

• The current version is not formatted according to Plos One.

We have now placed material and methods before results and discussion in the revised version.

• Please give due attention to all comments while revising your manuscript. I look forward to receive your revised submission.

All comments have been carefully incorporated in the revised version.

Reviewer #1: Overall, the manuscript entitled “Influence of soil moisture levels on the growth and reproductive behaviour of Avena fatua and Avena ludoviciana” presents novel and relevant results regarding the ability of these species in withstanding dryer soil conditions, a condition which is expected to become increasingly more common in the future due to ongoing global climate change.

Comment #1: It seems no information is presented regarding the actual cycle length across soil moisture levels. Is such information available?

We did not record the actual cycle length across soil moisture levels.

Comment #2: figures seem to be lacking quality, appearing blurred.

The figures have been revised again. 

Line 52: scientific names of two species which are referred to as “wild oats” throughout the manuscript had already been properly introduced (line 41-42); why inform their scientific names again? This can be confusing to readers.

Because both species are different and as per journal format, we preferred scientific names rather than common names throughout the manuscript. 

Line 321: this statement regarding the actual ability of crops such as wheat and barley seems vague and is not supported by any literature citation. It is clear that the authors are simply indicating this possibility, but it would be better to refer to articles (if any) which assessed these crops´ growth under water deficit. If none is found, then it seems necessary to add a sentence that “such statement remains to be evaluated” or similar.

The line has been modified as per the suggestion.

Reviewer #2: I have evaluated the manuscript "Influence of soil moisture levels on the growth and reproductive behaviour of Avena fatua and Avena ludoviciana" written by Sahil et al. Overall, it is an excellent piece of study and deserves publication.

I have a few minor suggestions (commented on the attached manuscript file).

All suggestions on the attached manuscript have been carefully incorporated.

The opening sentence of abstract defines climate change and study is on water stress. Only water stress can not be considered as climate change. Please rewrite the sentence.

The word “climate change” has been removed.

The Methods are given after results, while plos one requires methods before results.

The methods have been incorporated before the results now.

More info is needed on seed collection and storage. I suggest to add coordinates or a map showing the locations.

The required information about seed selection and storage has been added in the manuscript and coordinates have also been provided.

Authors have cited a survey work in Australia, could the current distribution map be provided in the manuscript?

This is a research paper and we think that a distribution map is not related to this study. That will be more suitable for a review paper or a book chapter.

I suggest to use common names of both species, how these are separated in Australia? The use of Latin names have been redundant throughout the manuscript. Obviously, common names will be redundantly used but better to use.

The common names of both species have been provided. In Australia, both species are known as wild/black oats; therefore, cannot differentiate. We have used common names used in the USA.

---

## [Editor Report · Decision Letter 1]

28 May 2020

PONE-D-20-06038R1

Influence of soil moisture levels on the growth and reproductive  behaviour of Avena fatua and Avena ludoviciana

PLOS ONE

Dear Dr. Mahajan,

Thank you for submitting your manuscript to PLOS ONE. After careful consideration, we feel that it has merit but does not fully meet PLOS ONE’s publication criteria as it currently stands. Therefore, we invite you to submit a revised version of the manuscript that addresses the points raised during the review process.

I have evaluated the revised manuscript. All the comments have been addressed. I have annotated the attached files for some minor changes. Please approve/address these changes before formal acceptance.

We look forward to receiving your revised manuscript.

Kind regards,

Shahid Farooq, Ph.D.

Academic Editor

PLOS ONE

---

## [Author Response · Author response to Decision Letter 1]

28 May 2020

Shahid Farooq, Ph.D.

Academic Editor

PLOS ONE

Subject: Ref.: PONE-D-20-06038 R1

Influence of soil moisture levels on the growth and reproductive behaviour of Avena fatua and Avena ludoviciana PLOS ONE

Dear Dr Shahid,

We thank you again for the critical look on the manuscript . The manuscript has been revised in the light of useful comments. All the comments have been incorporated in the revised version carefully. For clarity, the comments and suggestions will appear in the black colored text; while our response will appear in the blue text. In the manuscript also, specific changes can be seen in the track changed file. We hope that the revised manuscript is now acceptable in “PLOS ONE”.

We look forward to hearing from you in due course.

Sincerely yours

Gulshan Mahajan

Editor comments

 I have evaluated the revised manuscript. All the comments have been addressed. I have annotated the attached files for some minor changes. Please approve/address these changes before formal acceptance.

All your comments have been incorporated carefully and approved. In tables significant differences were differentiated with the help of letters.

---

## [Editor Report · Decision Letter 2]

1 Jun 2020

Influence of soil moisture levels on the growth and reproductive  behaviour of Avena fatua and Avena ludoviciana

PONE-D-20-06038R2

Dear Dr. Mahajan,

We are pleased to inform you that your manuscript has been judged scientifically suitable for publication and will be formally accepted for publication once it complies with all outstanding technical requirements.

With kind regards,

Shahid Farooq, Ph.D.

Academic Editor

PLOS ONE

Additional Editor Comments (optional):

The authors have addressed all the comments in the revised version of the manuscript. Therefore, current version can be accepted for publication.
---

## [Editor Report · Acceptance letter]

25 Jun 2020

PONE-D-20-06038R2 

Influence of soil moisture levels on the growth and reproductive  behaviour of Avena fatua and Avena ludoviciana 

Dear Dr. Mahajan:

I'm pleased to inform you that your manuscript has been deemed suitable for publication in PLOS ONE. Congratulations! Your manuscript is now with our production department. 

Kind regards, 

on behalf of

Dr. Shahid Farooq 

Academic Editor

PLOS ONE